# Epidemiological Survey of Four Reproductive Disorder Associated Viruses of Sows in Hunan Province during 2019–2021

**DOI:** 10.3390/vetsci9080425

**Published:** 2022-08-11

**Authors:** Qiwu Tang, Lingrui Ge, Shengguo Tan, Hai Zhang, Yu Yang, Lei Zhang, Zaofu Deng

**Affiliations:** 1Hunan Biological and Electromechanical Polytechnic, Changsha 410128, China; 2Animal Epidemic Prevention Station of Xiangxi Autonomous Prefecture, JiShou City 416000, China; 3Animal Disease Prevention and Control Center, Wangcheng District, Changsha 410128, China; 4Subdistrict Office of Nanzhuangping Street, Yongding District, Zhangjiajie 427000, China

**Keywords:** sow, reproductive disorder associated viruses, epidemiology, genotype diversity, Hunan province

## Abstract

**Simple Summary:**

Porcine reproductive and respiratory syndrome virus (PRRSV), porcine circovirus type 2 (PCV2), pseudorabies virus (PRV), and classical swine fever virus (CSFV) are major causative agents contributing to the reproductive disorders in sows in China. This is the first research investigating the epidemiology and molecular characteristics of these four viruses in sows with reproductive disorders in China. In this study, 407 aborted-fetus samples were obtained from 89 pig farms in Hunan province from 2019 to 2021 for molecular detection and phylogenetic analysis of PRRSV, PRV, PCV2, and CSFV. Collectively, various patterns of singular infections or co-infections were identified in pig herds; PRRSV and PCV2, especially, were frequently detected in the collected specimens. Further phylogenetic analysis revealed the complex genotypes of each virus prevalent in pig populations in Hunan province. In conclusion, this study provides the latest information on the epidemiology and genotype diversity of four viruses in sows with reproductive diseases in Hunan province of China, which would contribute to developing effective strategies for disease control.

**Abstract:**

Porcine reproductive disorders have been considered as the major factors that threaten pig industries worldwide. In this study, 407 aborted-fetus samples were obtained from 89 pig farms in Hunan province, to investigate the prevalence of four viruses associated with porcine reproductive disease, including porcine reproductive and respiratory syndrome virus (PRRSV), porcine circovirus type 2 (PCV2), pseudorabies virus (PRV), and classical swine fever virus (CSFV). Meanwhile, the target gene sequences of representative PRRSV (ORF5), PCV2 (ORF2), CSFV (E2), and PRV (gE) strains were amplified, sequenced, and analyzed. The results showed that the positive rates of PRRSV, PCV2, PRV, and CSFV among the collected samples were 26.29% (107/407), 52.83% (215/407), 6.39% (26/407), and 12.29% (50/407), respectively. Moreover, co-infection with two and three pathogens were frequently identified, with PCV2/PRRSV, PRRSV/CSFV, PRRSV/PRV, PCV2/CSFV, PCV2/PRV, and PRRSV/PCV2/CSFV mix infection rates of 9.09%, 3.19%, 2.95%, 3.69%, 2.21%, and 0.49%, respectively. Moreover, ORF5-based phylogenetic analysis showed that 9, 4, and 24 of 37 PRRSV strains belonged to the PRRSV2 lineages 1, 5, and 8, respectively. ORF2-based phylogenetic analysis revealed that PCV2d and PCV2b were prevalent in Hunan province, with the proportions of 87.5% (21/24) and 12.5% (3/24), respectively. An E2-based phylogenetic tree showed that all 13 CSFV strains were clustered with 2.1 subgenotypes, these isolates were composed of 2.1b (10/13) and 2.1c (3/13) sub-subgenotypes. A gE-based phylogenetic tree showed that all six PRV strains belonged to the genotype II, which were genetically closer to variant PRV strains. Collectively, the present study provides the latest information on the epidemiology and genotype diversity of four viruses in sows with reproductive diseases in Hunan province, China, which would contribute to developing effective strategies for disease control.

## 1. Introduction

In recent years, the pig industry has rapidly developed in China with a large-scale breeding scale proportion and frequent transportation of livestock. However, reproductive disorders in sows remain a major health concern threatening the pig industry in China, which are mainly characterized by high fever, abortion, stillborn fetus, etc. A variety of pathogens are capable of causing reproductive disorders in sows; among them, porcine reproductive and respiratory syndrome virus (PRRSV), porcine circovirus type 2 (PCV2), classical swine fever virus (CSFV), and pseudorabies virus (PRV) have been widely prevalent in Chinese pig populations, and are the predominant viral agents leading to reproductive failure in breeding pigs and causing huge economic losses [1,2,3,4].

PRRSV is an enveloped positive-sense RNA virus that belongs to the *Arteriviridae* family [5]. The PRRSV genome is nearly 15 kb in length, which contains more than 10 open reading frames (ORFs). According to their genetic features, PRRSVs are divided into two genotypes (PRRSV1 and PRRSV2). PRRSV2 isolates are mainly prevalent in Chinese swine herds, which are further divided into four lineages (lineages 1, 3, 5 and 8) [6]. PCV2 belongs to the family *Circoviridae* which has a circular single-stranded DNA genome of ~1.7 kb [7]. The PCV2 genome contains three ORFs, and PCV2 strains which are currently prevalent worldwide can be divided into eight genotypes (PCV2a to PCV2h) based on the nucleotide diversity of the ORF2 gene [7]. CSFV, the member of genus *Pestivirus*, is a single-stranded positive-sense RNA virus, and the CSFV genome (~12.3 kb) contains only a large ORF encoding a polyprotein precursor of 3898 amino acids [8]. Based on the genetic diversity of the E2 gene, Chinese CSFV isolates are divided into four subgenotypes (1.1, 2.1, 2.2 and 2.3), and the subgenotype 2.1 has been predominant in China in recent years [9]. PRV belongs to the genus *Varicellovirus* of the family *Herpesviridae*, which is an enveloped and double-stranded liner DNA virus [3]. The genome of PRV is approximately 143 kbp encoding more than 70 ORFs [3]. PRV strains are classified into two genotypes (genotype I and genotype II), and the potential threat of PRV variants in genotype II to humans’ health has drawn wide attention recently [10].

PRRSV, PCV2, CSFV, and PRV are important pathogens causing reproductive disorders in sows. Additionally, these viruses have been widely prevalent in pig populations in China, and co-infections of two or more pathogens are often observed [2,11,12]. However, the prevalence and co-infection characteristics of PRRSV, PCV2, CSFV, and PRV in sows with reproductive disorders in China are rarely studied. The main objective of this research was to investigate the epidemiology and co-infection status of these four viruses in reproductive-disorder-related sows in Hunan province from 2019 to 2021. Moreover, the genetic features of the representative PRRSV, PCV2, PRV, and CSFV strains were also investigated. This information will shed light on taking efficient measures to control reproductive disorders in sows in China.

## 2. Materials and Methods

### 2.1. Sample Collection and Pre-Treatment

From April 2019 to December 2021, aborted-fetus samples (lymph node, lung, and kidney, etc.) were collected from 407 sows with reproductive diseases (e.g., stillbirths, abortion, mummified fetuses) from 89 pig farms in 12 cities, nearly covering the entire Hunan province, and these samples were sent to Hunan Biological and Electromechanical Polytechnic for further processing. Each sample was mixed with sterilized physiological saline, homogenized, frozen and thawed three times. The supernatants were collected after being centrifuged, and stored in a −80 °C fridge.

### 2.2. Nucleic Acid Extraction and Pathogen Detection

Viral RNA and DNA were extracted from each collected specimen using the MiniBEST Viral DNA/RNA Extraction kit Ver.5.0 (TaKaRa, Beijing, China) according to its operating instructions. The presence of PCV2 (targeting the ORF1 gene), PRV (targeting the gE gene) and CSFV (targeting the E2 gene), PRRSV (targeting the ORF5 gene) was detected by PCR or reverse transcription (RT)-PCR. The primers used for pathogen identification in this study were shown in Table 1. The positive PCR products were defined to generate the anticipated DNA bands in 2% agarose gel electrophoresis.

### 2.3. Cloning and Sequencing

The full lengths of ORF2 (PCV2), ORF5 (PRRSV), gE (PRV), and E2 (CSFV) of the representative strains were amplified using four pairs of primers as shown in Table 1. The positive PCR products were sent to a commercial company for sequencing in duplicate. The targeting sequences of different strains were submitted to NCBI database (Appendix A).

### 2.4. Bioinformatics Analysis

To investigate the genetic evolutions of PRRSV, PCV2, CSFV, and PRV strains obtained in this study, phylogenetic trees were constructed based on the corresponding genes using the neighbor-joining method in MEGA 7.0 software [Kimura 2-parameter, 1000 bootstrap replicates]. The reference strains used in the present study were shown in the Appendix A.

## 3. Results

### 3.1. The Epidemiology of These Four Pathogens in Clinical Samples

As shown in Table 2, at the herd level, clinical samples of 20 (22.47%), 34 (38.20%), 13 (14.61%), and 2 (2.25%) pig farms were singularly infected with PRRSV, PCV2, CSFV, and PRV, respectively. Dual infections were observed in 14 (15.73%), 4 (4.49%), 5 (5.62%), 6 (6.74%), and 4 (4.49%) pig farms, where PRRSV + PCV2 co-infection, PRRSV + CSFV co-infection, PRRSV + PRV co-infection, and PCV2 + CSFV co-infection were identified, respectively. Furthermore, triple-pathogen infections were only observed in one pig herd.

At the sample level, 26.29% (107/407), 52.83% (130/407), 12.29% (50/407), and 6.39% (26/407) samples were positive for PRRSV, PCV2, CSFV, and PRV, respectively (Table 2). In total, 43 (10.57%), 67, 20, and 5 tissue samples were solely positive for PRRSV, PCV2, CSFV, and PRV, respectively. As for the co-infection status, the co-infection of PRRSV and PCV2 among the collected tissue samples was often identified, with the highest proportion of 9.09% (37/407). The proportions of simultaneous infection of PRRSV and CSFV, PRRSV and PRV, PCV2 and CSFV, PCV2 and PRV were 3.19% (12/407), 2.95% (12/407), 3.69% (15/407), and 2.21% (9/407). Furthermore, only two pigs were concurrently infected with PRRSV, PCV2, and CSFV. Collectively, 33.17% (135/407), 21.13% (86/407), and 0.49% (2/407) samples were single infection, dual infection, and triple infection, respectively.

### 3.2. Phylogenetic Analysis of PRRSV Strains

To investigate the genetic characteristics of PPRSV strains obtained in this study, the ORF5 genes of 37 randomly selected PPRSV-positive samples were amplified, sequenced, and analyzed. The ORF5 gene-based phylogenetic analysis showed that all 37 PRRSV strains collected in the present study belonged to the North American genotype (PRRSV2) and they were divided into three lineages: lineage 1 (NADC30-like), lineage 5 (VR2332-like), and lineage 8 (JXA1-like), the proportions of which were 29.32% (9/37), 10.81% (4/37), and 64.86% (24/37), respectively (Figure 1).

### 3.3. Phylogenetic Analysis of PCV2 Strains

A phylogenetic tree was generated based on the 24 PCV2 ORF2 gene sequences collected in the present study and 12 PCV2 reference strains with different genotypes (Figure 2). The results revealed that 21 PCV2 ORF2 sequences were clustered with PCV2d strains, and the others (three strains) and the representative PCV2b strains were grouped in the same branch, while other genotypes (such as PCV2a and PCV2c) were not identified in this study.

### 3.4. Phylogenetic Analysis of CSFV Strains

In the present study, a total of 13 CSFV E2 gene sequences were successfully amplified (1119 nt in length); the phylogenetic tree was generated based on the E2 sequences of 13 CSFV strains obtained here and 22 CSFV reference strains obtained from GenBank database (Figure 3). The results showed that all CSFV strains collected in this study belonged to the subgenotype 2.1. Among which, 10 and 3 CSFV strains belonged to the sub-subgenotype 2.1b and sub-subgenotype 2.1c, respectively.

### 3.5. Phylogenetic Analysis of PRV Strains

Phylogenetic analysis was performed based on the gE sequences of 6 PRV strains obtained in this study and 16 PRV reference strains downloaded from GenBank database (Figure 4). All PRV strains could be divided into two genotypes, namely genotype I and genotype II. Nearly all Chinese PRV strains belonged to the genotype II, which were composed of classical PRV strains (genotype II-a) and variant PRV strains (genotype II-b). Notably, six PRV strains collected here showed closer genetic relationships with PRV variants in genotype II-b compared with others.

## 4. Discussion

China has the largest pig population in the world, while viral porcine reproductive disorders are considered as major factors threatening the pig industry in China. PRRSV, PCV2, PRV, and CSFV are widely prevalent in the Chinese pig population, and the clinical symptoms of the disease caused by any of these four viruses include porcine reproductive diseases, such as abortion, stillbirth, etc.. The co-prevalences of these viruses are often observed in Chinese pig farms, which may contribute to the fact that co-infection are more frequently observed than single infection [9,12]. Moreover, the co-infection of multiple viruses (such as PRRSV, CSFV, and PCV2) could cause more severe clinical symptoms and death rate than a singular infection, and also cause heavy damage to the host immunization efficacy [13,14]. Thus, monitoring the co-infection status and investigating the molecular characteristics of these four viruses (PRRSV, PCV2, CSFV, and PRV) are beneficial for the control and prevention of viral reproductive disorders in the pig industry. Additionally, this will accelerate the development of novel vaccines for the corresponding pathogen prevention.

In this study, the co-infectious status of four viruses (PRRSV, PCV2, CSFV, and PRV) in aborted-fetus samples from Hunan province in recent years was evaluated and several characteristics were summarized: (A) 54.79% (233/407) samples obtained in this study were PCR-positive for at least one pathogen, and the average detection rates of PRRSV, PCV2, CSFV, and PRV were 26.29% (107/407), 31.94% (130/407), 12.29% (50/407), and 6.39% (26/407), respectively. In Chen’s research, the PCV2 and PRRSV nucleic acids were also detected more frequently than CSFV and PCV3 in diseased pigs [9]. Moreover, the above results indicated that other reproductive-disorder-related pathogens might be prevalent in the investigated pig farms, such as parasites (*Toxoplasma gondii*), bacteria (such as *Brucella* and *Chlamydia*), and other viruses (Porcine parvovirus and Japanese encephalitis virus, etc.), which needs further investigation. (B) Five types of double infections were observed, in which the detection rates ranged from 2.21% (PCV2+PRV) to 9.9% (PCV2+PRRSV). Owing to the severe prevalence of PCV2 and PRRSV in the Chinese pig industry, the co-infection of PCV2 and PRRSV was often identified with the positive rates varying between 10.69% and 52.42% in previous research [8,15,16]. Collectively, the above results suggest that these four viruses remained the major threats affecting the pig industry in Hunan province. Co-infection in different combinations was frequently observed; thus, it is necessary to develop molecular methods for differential detection of multiple viruses, and the co-infection of multiple viruses in clinical samples should be concerned.

Multiple lineages of PRRSV2 including lineages 1, 3, 5, and 8 have been identified in China. However, the lineage 8, especially sublineage 8.7, has been predominant in Chinese pig populations [6,17,18]. Phylogenetic analysis showed that all PRRSV strains obtained in this study were grouped with PRRSV2, which could be further divided into three lineages (1, 5, 8). Moreover, 24 of out 37 (64.86%) the strains belonged to the lineage 8 (genetically close to JXA1 and HuN strains). The highly pathogenic PRRSV MLV vaccines have been widely applied in Chinese pig herds [6]. Meanwhile, the virulence reversion of attenuated vaccine strains was often observed [7,19]. These factors might contribute to the high prevalence of PRRSV lineage 8 in Hunan province. In addition, the present results also revealed the prevalence of PRRSV NADC30-like strains in China, although its proportion was relatively lower compared with the lineage, growing evidence showed that MLV vaccines could not provide efficient cross-protection to NADC30-like strains [20], which suggested that biosecurity improvement but not the application of vaccines should be the priority for PRRS control.

Currently, PCV2 strains are composed of eight genotypes (PCV2a-PCV2h) [21]. In China, the genotype switching from PCV2b to PCV2d has been observed since 2012 [7]. Qu et al. revealed that PCV2d has become predominant in Hunan province after 2008 [22]; consistent with these observations, our results also showed that all PCV2 strains identified in this study were composed of PCV2b and PCV2d genotypes, and the PCV2d should be the predominant genotype with a proportion of 87.5% (21/24).

Different subgenotypes (2.1, 2.2, and 1.1) of CSFV have been prevalent in Chinese pig herds, among which 2.1 was considered as the predominant subgenotype in China. Moreover, the subgenotype 2.1 strains could be further divided into four sub-subgenotypes (2.1a, 2.1b, 2.1c, and 2.1d). However, the predominant sub-subgenotypes which were prevalent in different regions in China were diverse [9,23]. In this study, phylogenetic analysis showed that both sub-subgenotypes 2.1b and 2.1c were prevalent in pig herds in Hunan province, and the proportion of the former one (76.92%, 10/13) was higher than the latter one (23.08%, 3/13). These observations suggested that the sub-subgenotype 2.1b should be dominant in Hunan province, which was inconsistent with that in Guangdong province (2.1c) [23].

PRV strains prevalent in China could be divided into two genotypes (genotype I and genotype II); the genotype II strains were composed of classical PRV strains and variant PRV strains, which were prevalent in China before 2011 and after 2011, respectively [3]. Additionally, recently, PRV variant strains have been the predominant subgenotype in China, and accumulating evidence confirmed the cross-species transmission of PRV variants from animals to humans [3]. Notably, all six PRV strains obtained here showed closer genetic relationships with PRV variants compared with others, suggesting the persistent threats of PRV variants to pig herds in Hunan province. Moreover, one case of variant PRV infection in human was recently documented in Hunan province [24], which indicated the potential threats of PRV variants to humans, especially the workers who are engaged in the pig industry, should be concerned.

In summary, the co-infection situation and genetic characteristics of four viral pathogens in sows with reproductive disorders in Hunan province from 2019 to 2021 were investigated. Collectively, various patterns of singular infections or co-infections were identified in pig herds; especially, PRRSV and PCV2 were frequently detected in the collected specimens. Further phylogenetic analysis revealed the prevalent genotypes of each virus in pig populations in Hunan province. These findings have provided a better understanding of the epidemiology of these four viruses in sows with reproductive disorders. Thus, it is essential to carry out laboratory diagnosis in real cases (not only reproductive disorders in sows), which will be beneficial for disease prevention and control.

## Figures and Tables

**Figure 1 vetsci-09-00425-f001:**
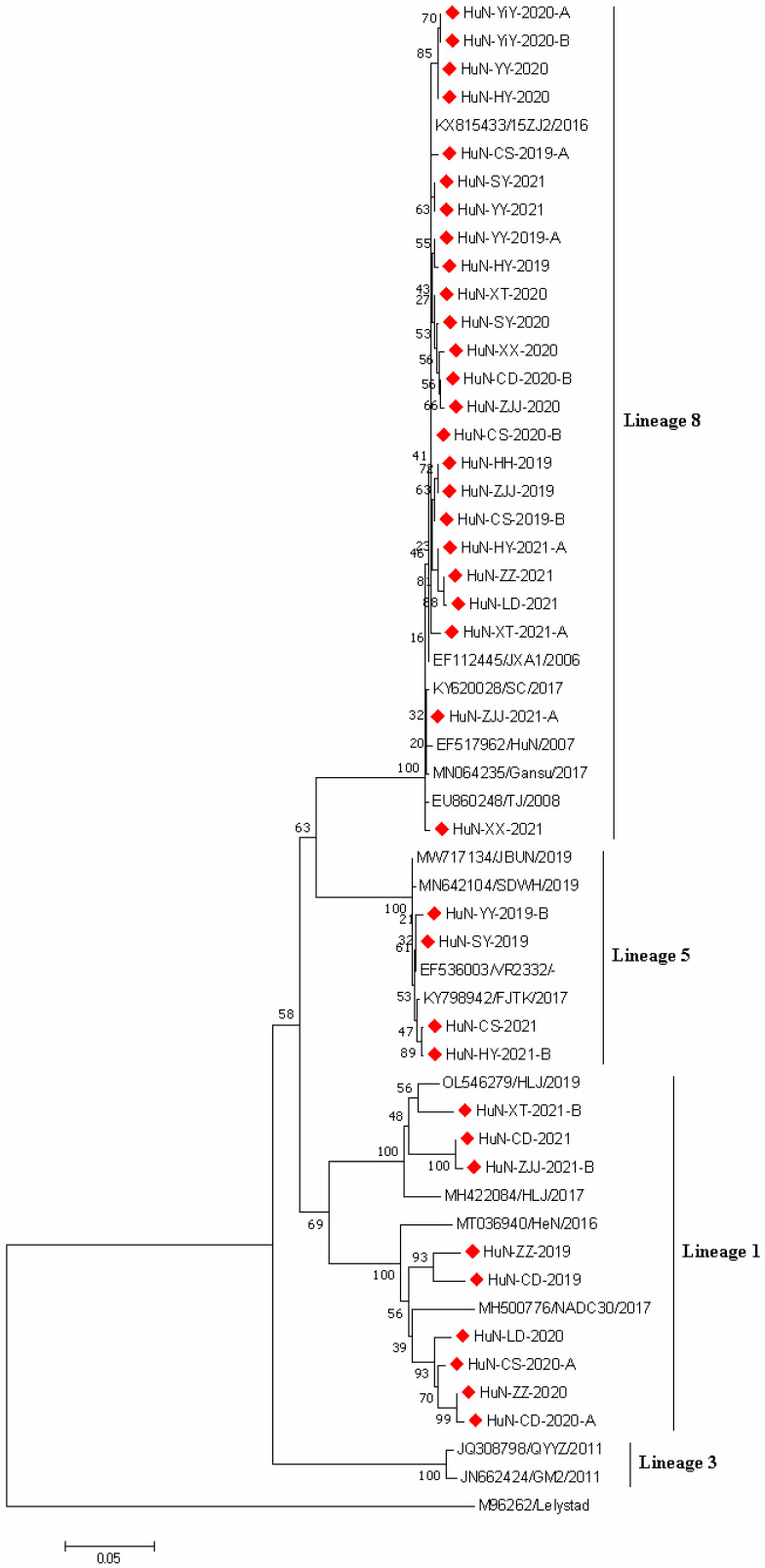
Phylogenetic tree based on *ORF5* gene sequences of 37 PRRSV strains identified in this study and other reference strains generated by the neighbor-joining method in MEGA 7.0 software, with 1000 bootstrap support. The red prismatics represent these newly identified PRRSV strains.

**Figure 2 vetsci-09-00425-f002:**
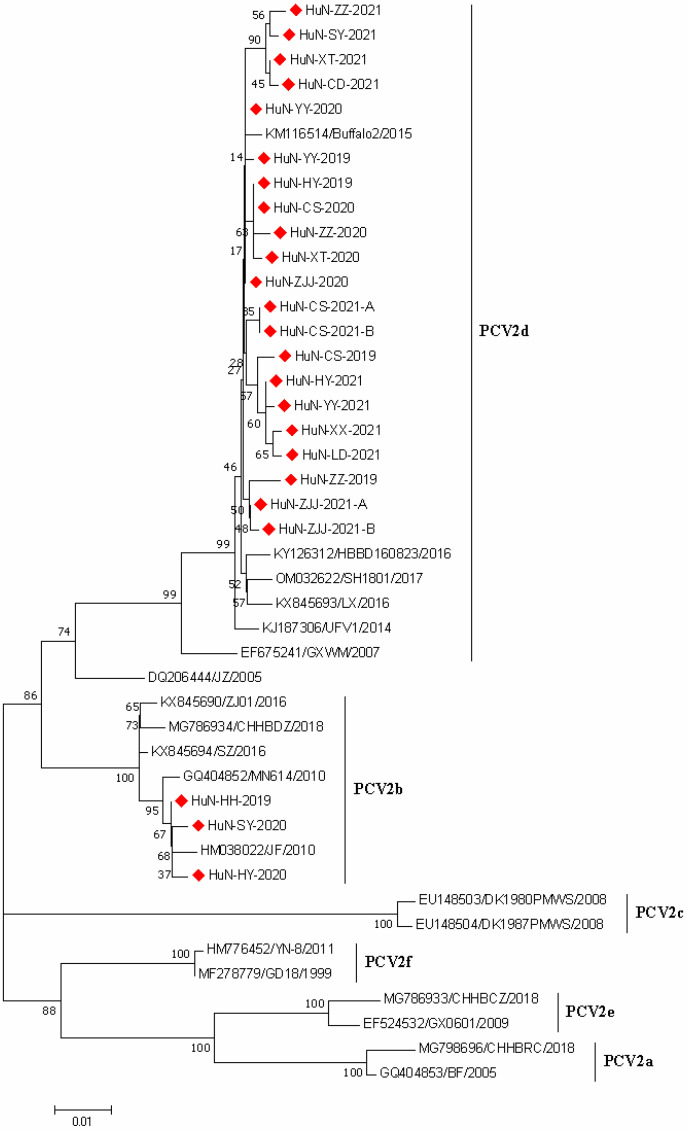
Phylogenetic tree based on ORF2 gene sequences of 24 PCV2 strains identified in this study and other reference strains generated by the neighbor-joining method in MEGA 7.0 software, with 1000 bootstrap support. The red prismatics represent these newly identified PCV2 strains.

**Figure 3 vetsci-09-00425-f003:**
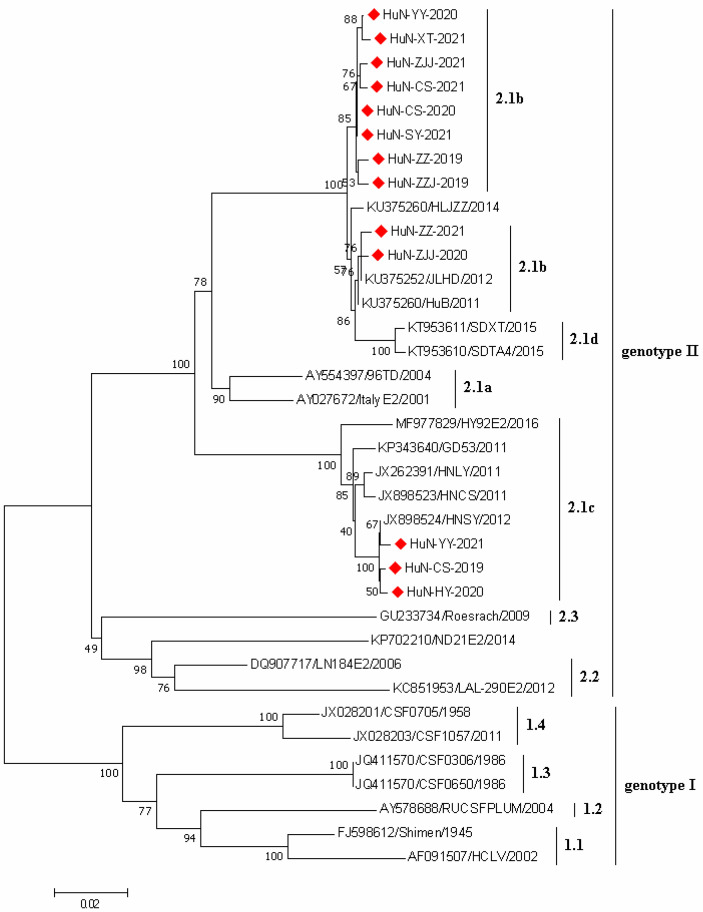
Phylogenetic tree based on E2 gene sequences of 13 CSFV strains identified in this study and other reference strains generated by the neighbor-joining method in MEGA 7.0 software, with 1000 bootstrap support. The red prismatics represented these newly identified CSFV strains.

**Figure 4 vetsci-09-00425-f004:**
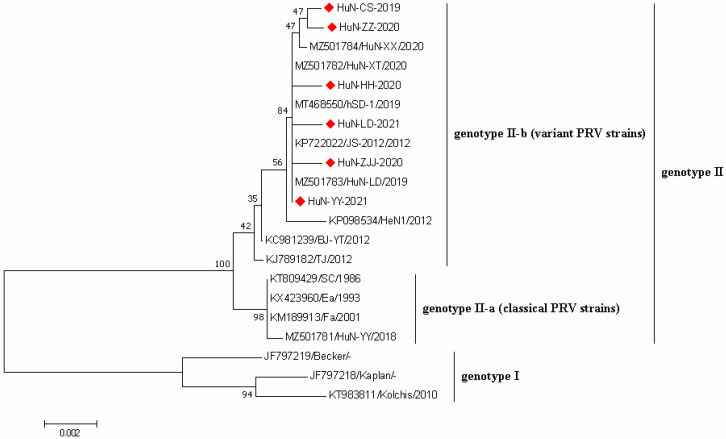
Phylogenetic tree based on gE gene sequences of 6 PRV strains identified in this study and other reference strains generated by the neighbor-joining method in MEGA 7.0 software, with 1000 bootstrap support. The red prismatics represented these newly identified PRV strains.

**Table 1 vetsci-09-00425-t001:** Primers used in this study.

Primer	Sequence (5′-3′)	Binding Position	Length	Purpose	Reference Sequence
PRRSV-ORF5-F	CAACCGTTTTAGCCTGTCTT	13734–13753	709 bp	Detection of PRRSV and Sequencing	PRU87392
PRRSV-ORF5-R	CAAAACGCCAAAAGCACC	14425–14442			
PCV2-OFR2-F	CCATGCCCTGAATTTCCATATGAAAT	960–985	850 bp	Detection of PCV2 and Sequencing	MH191375
PCV2-OFR2-R	TGAGGTGCTGCCCAGGTGCT	23–42			
CSFV-NS5B-F	CCTTAACCATGCACATGTCAG	11049–11069	574 bp	Detection of CSFV	FJ529205
CSFV-NS5B-R	TCAGTTGACAACACCAATAAG	11602–11622			
CSFV-E2-F	GTAAATATGTGTGTGTTAGACCAGA	2211–2235	1402 bp	Sequencing	FJ529205
CSFV-E2-F	GTGTGGGTAATTGAGTTCCCTATCA	3588–3612			
PRV-dgE-F	CCGAGTACGTCACGGTCATC	126232–126251	555 bp	Detection of PRV	KP257591
PRV-dgE-R	CTTCCGGTTTCTCCGGATCG	126767–126766			
PRV-gE-F	ATGCGGCCCTTTCTGCTGCGC	125108–125128	1740 bp	Sequencing	KP257591
PRV-gE-R	TTAAGCGGGGCGGGACATCAAC	126826–126847			

**Table 2 vetsci-09-00425-t002:** Infection status of four viruses in 407 aborted-fetus samples collected from 2019 to 2021 in Hunan province.

Infection Categories	Virus	Infection Status in Surveyed Pig Farms (*n* = 89)		Infection Status in Collected Pig Samples (*n* = 407)		
		Positive Farms	Percentage (%)	Positive Samples	Percentage (%)	Subtotal (%)
Single infection	PRRSV	20	22.47	43	10.57	33.17
	PCV2	34	38.20	67	16.46	
	CSFV	13	14.61	20	4.91	
	PRV	2	2.25	5	1.23	
Dual infection	PRRSV + PCV2	14	15.73	37	9.09	21.13
	PRRSV + CSFV	4	4.49	13	3.19	
	PRRSV + PRV	5	5.62	12	2.95	
	PCV2 + CSFV	6	6.74	15	3.69	
	PCV2 + PRV	4	4.49	9	2.21	
Triple infection	PRRSV + PCV2 + CSFV	1	1.12	2	0.49	0.49
Total	PRRSV	-	-	107	26.29	54.79
	PCV2	-	-	130	31.94	
	CSFV	-	-	50	12.29	
	PRV	-	-	26	6.39	

## Data Availability

All datasets generated in this study are available in the published article and its Appendix A.

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
