# Peer review of "Epidemiological Survey of Four Reproductive Disorder Associated Viruses of Sows in Hunan Province during 2019–2021"

_vetsci, 2022, doi:10.3390/vetsci9080425_

Round 1
Reviewer 1 Report
In this scientific paper, the authors pointed out the presence of virus infection and co-infection in aborted fetuses of pigs. Specifically, porcine reproductive and respiratory syndrome virus (PRRSV) and porcine circovirus type 2 (PCV2) were found to be present.
In the supplementary materials section, only the supplementary tables can be downloaded. In the "results" section, figures and tables are mentioned but cannot be seen.
Please upload the complete results.
Regarding the results I have a few questions:
1) is it possible to trace the weeks of gestation of the aborted fetuses?
2) what kind of reproductive-related problems did the sows have? Exclusively related to virus infections? Please specify if this is possible in the text.
Overall, the following study could have impact in improving epidemiological conditions in pig farms.
Author Response
In the supplementary materials section, only the supplementary tables can be downloaded. In the "results" section, figures and tables are mentioned but cannot be seen.
Answer: We are sorry for this, and the figures and tables would be available in the revised manuscript. Tables were listed at the end of revised manuscript, and the figures were listed as enclosures.
Regarding the results I have a few questions:
- is it possible to trace the weeks of gestation of the aborted fetuses?
Answer: Thanks for your consideration. However, we are sorry that we have no access to trace the weeks of gestation of the aborted fetuses. Firstly, we did not realize the significance of these information before the investigation carried out, and now it is too late. Secondly, most of pig farmers were unwilling to provided these information when we asked them for help.
- A) what kind of reproductive-related problems did the sows have? Exclusively related to virus infections? Please specify if this is possible in the text.
Answer: Thanks for your consideration in this section. A) the clinical symptoms of reproductive-related sow mainly include stillbirths, abortion, mummified fetuses,etc., and these information have been added in the revised manuscript. 2) we totally agree with it that other factors could lead to reproductive disorders in sows exception for viral infection, such as bacterial and parasitic infections. We have added these description in the section of discussion.
- Overall, the following study could have impact in improving epidemiological conditions in pig farms.
Answer: Thank you very much for your comments.

Reviewer 2 Report
The manuscript “Epidemiological survey of four porcine reproductive disorder 2 associated viruses of sows in Hunan Province during 2019-2021” by Tang et al. provided information on the epidemiology and genotype diversity of four viruses in sows with reproductive diseases in Hunan Province - China. The question addressed in this manuscript can be of interest for the scientific community. Suggestions follow for the authors.
Title: Please delete “porcine”.
Introduction: The objective is not sufficiently clear. It needs to be re-written.
Materials and methods: Would an ethical approval number be required for research on animal samples?
Results: This reviewer did not have access to figures and tables. Please provide them.
Discussion: I think part of the discussion needs improvement. I consider the lack of some lines that give greater importance to the work carried out (that highlight it), emphasizing its biological relevance and possible applications of the results.
Author Response
The manuscript “Epidemiological survey of four porcine reproductive disorder 2 associated viruses of sows in Hunan Province during 2019-2021” by Tang et al. provided information on the epidemiology and genotype diversity of four viruses in sows with reproductive diseases in Hunan Province - China. The question addressed in this manuscript can be of interest for the scientific community. Suggestions follow for the authors.
Title: Please delete “porcine”.
Answer: Thanks for your suggestion, we have deleted it.
Introduction: The objective is not sufficiently clear. It needs to be re-written.
Answer: Thanks a lot for your suggestion, the objective has been re-written in the revised manuscript, we hope it would be suitable for publication.
Materials and methods: Would an ethical approval number be required for research on animal samples?
Answer: Thank for your consideration in this section. We have provided the ethical approval statement in the revised manuscript.
Results: This reviewer did not have access to figures and tables. Please provide them.
Answer: We are sorry for this, and the figures and tables would be available in the revised manuscript. Tables were listed at the end of revised manuscript, and the figures were listed as enclosures.
Discussion: I think part of the discussion needs improvement. I consider the lack of some lines that give greater importance to the work carried out (that highlight it), emphasizing its biological relevance and possible applications of the results.
Answer: Thank you very much for your suggestion. We have added the importance description in the section of discussion. We hope which will be suitable.

Round 2
Reviewer 2 Report
This is a 2nd submission of the manuscript titled "Epidemiological survey of four porcine reproductive disorder 2 associated viruses of sows in Hunan province during 2019-2021". The authors addressed most of the concerns raised by the reviewer. However, figures are mentioned in results sections but cannot be seen. Again, please upload the complete results.
Furthermore, minor suggestions follow for the authors:
Line 68: “Meanwhile, the…”
Line 76: In “…abortion, mummified fetuses.)”, delete “.” (dot).
Line 86: “…reverse transcription (RT)-PCR. The primers used…”
Lines 93 and 94: “Supplementary Tables 1-4”.
Line 100: “Supplementary Tables 1-4”.
Line 103: Table 1? Is this correct?
Lines 173 and 174: In “…Japanese encephalitis virus, etc.,)…”, delete “,” (comma after etc.).
Line 195: “…the biosecurity…”
Line 232: “…which would…”
All text: "Province" or "province"? Standardize it.
Author Response
This is a 2nd submission of the manuscript titled "Epidemiological survey of four porcine reproductive disorder 2 associated viruses of sows in Hunan province during 2019-2021". The authors addressed most of the concerns raised by the reviewer. However, figures are mentioned in results sections but cannot be seen. Again, please upload the complete results.
Reply: Thanks for your comments here, we are so sorry that the figures are not available in the manuscript. Here, we have inserted figures into the revised manuscript, so that you can consult them.
Furthermore, minor suggestions follow for the authors:
Line 68: “Meanwhile, the…”
Rely: Thanks for your consideration here, we have changed the word “meanwhile” with “moreover”.
Line 76: In “…abortion, mummified fetuses.)”, delete “.” (dot).
Reply: Thanks a lot, we have modified it according to your correction.
Line 86: “…reverse transcription (RT)-PCR. The primers used…”
Reply: Thanks a lot, we have modified it.
Lines 93 and 94: “Supplementary Tables 1-4”.
Reply: Thanks a lot, we have modified it.
Line 100: “Supplementary Tables 1-4”.
Reply: Thanks a lot, we have modified it.
Line 103: Table 1? Is this correct?
Reply: we are sorry for this, it should be table 2, but not table, and we have corrected it in the revised manuscript.
Lines 173 and 174: In “…Japanese encephalitis virus, etc.,)…”, delete “,” (comma after etc.).
Reply: Thanks a lot, we have corrected it.
Line 195: “…the biosecurity…”
Reply: Thanks a lot, we have corrected it.
Line 232: “…which would…”
Reply: Thanks a lot, we have changed the “would” with “will”, which lead to the consistent tense.
All text: "Province" or "province"? Standardize it.
Reply: Thanks for your consideration in this section, the word “province” has been standardized.
